# A Method to Improve the Design of Virtual Reality Games in Healthcare Applied to Increase Physical Activity in Patients with Type 2 Diabetes Mellitus

Leticia Neira-Tovar [1,*], Iván Castilla Rodríguez [1] and Francisco González Salazar [2]

1   I3MA Doctoral Program, Department of Computer Engineering and Systems, Universidad de La Laguna, S/C de Tenerife, 38200 San Cristóbal de La Laguna, Spain
2   Health Sciences Department, Universidad de Monterrey, Monterrey 66238, Mexico
*   Correspondence: leticia.neira@gmail.com

**Abstract:** In recent years, there has been an increase in research related to specific applications or features of virtual reality tools for health care. Each work highlights their findings related to the specific case on which the research was focusing. The present study is a proposal of a design method for virtual reality games for healthcare applications. Virtual reality allows the integration of real physical activity, required for health treatments, and a virtual world that improves the engagement of the patient. The design method is based on the needs that a health treatment faces in terms of motivational activities and challenges to include in a virtual reality scenario, obtained from the investigation of previous works. A treatment benefiting from user exercise was selected as the case to fully describe the design method. Therefore, this paper also describes a virtual reality tool developed to help in the treatment of chronic diseases, such as type 2 diabetes. The tool was used by a group of patients. To evaluate the method and the virtual reality application, two types of tests were applied: a clinical test adapted from previous literature and a usability test focused on the virtual environment. Clinical tests showed low stress and good physical health, while usability tests showed engagement.

**Keywords:** design games; virtual reality; health games; diabetes physical treatments; immersive reality





## 1. Introduction

In recent years, the prevalence of diabetes has been increasing globally, especially in low- and middle-income countries. This rise in prevalence is related to modifiable risk factors, such as alimentation habits, physical activity, and overweight and obesity. Although the causes of physical damage to the motor activity of the patient have not yet been completely determined, they are usually associated with poor disease control, adverse health outcomes, and quality of life impairment [1]. Therefore, much of the diabetes burden can be prevented by behavioral changes that include healthy food and favoring regular physical activity [2].

New methods are being created for the prevention and rehabilitation of some of the negative physical and psychological consequences of diabetes. Technological progress has led to more sophisticated technology for diabetes prevention [3], as well as diabetes monitoring [4]. Among the different technologies applied to healthcare, virtual reality has been successfully employed to prevent and educate people on the consequences of chronic diseases, such as diabetes [5]. The use of serious games is another example [6]; games can be designed as prototypes to teach healthy lifestyle habits [7] and to perform physical activity [1], mainly in diabetic patients of a single community [8]. Additionally, gamification is a very useful tool to motivate people to prevent this disease and its consequences by means of the continuous use of fun and healthy products [9].

Video games have demonstrated benefits on improving physical activity and health treatments [10,11]. The use of virtual reality tools can benefit people of any age: from

children in a rehabilitation process [12] to older adults who need to improve balance [13,14]. Nevertheless, some studies have pointed out how usability together with quality do not fully symbolize player satisfaction [15]. Therefore, playability depends not only on the experience of users but also on the characteristics of the patient condition. Specialists on preventive health games are working hard to develop better and useful tools [16], and some of them include clinical tests to validate their product. Even though there are some related contributions aimed to design rehabilitation games [17], there are still some areas of opportunity to design games for health that use a virtual or mixed reality tool, which may be summarized by the following questions: what about product validation tests during the patient interaction? How to test the game product to know whether it fulfills the minimum patient requirements, such as product satisfaction? Is the product capable of achieving the rehabilitation target? Such questions motivate this research work, which seeks to answer them in a comprehensive way. Hence, a new method is proposed in which the objective is to help in the design of mixed reality applications to be used in health treatments. Starting with a review of related works, it was summarized that digital game-based learning and serious games are receiving more attention [18], so testing them before being used in health activities is quite important. Although the number of evaluations is constantly increasing [16,17], there are still some weaknesses. One of the questions is what the requirements are for a good evaluation framework for serious game research; the work of Mayer [19] enumerates the requirements for a good evaluation framework for serious games: broad in scope, comparative, standardized, specific, flexible, triangulated, multileveled, validated, expandable, unobtrusive, fast and non-time consuming, and multi-purposed. Clients are not usually interested in evaluating beyond their own immediate needs, and there usually are case-specific evaluation questions [20,21]. These situations stand in the way of comparative, longitudinal research.

Diabetes has started to become a study case in previous virtual reality research. Diabetes is one of the most prevalent diseases in the world, associated with great morbidity and mortality, and the main tools for controlling this disease are diet and exercise. The development of this method can offer millions of users the possibility of exercising from home with suitable technological tools. This is of great significance now, in times of confinement where the possibilities of exercising outdoors are reduced even in older people. Previous work using virtual reality has employed a wide range of approaches to improve the habits of a patient. On one hand, some of the studies have focused on improving healthy eating habits [5]. On the other hand, some other studies have considered that physical exercise is key to improving the health of the patient [7]. Some researchers have worked on investigating and examining whether exposure to greater active videogame variety increases [22].

Other authors have worked on how to introduce concepts, such as competitive effort, perceived difficulty, and connected gameplay, and ways to develop a competitive gameplay must be designed to facilitate learning [23].

While researching related works as background for this paper, a general architecture for the development of education web portal research was found, unlike this proposal method, in which one support performance evaluation along the sequence of activities was composed of serious games and virtual environments [24], where the objectives were divided into three domains: cognitive, affective, or psychomotor. However, what about the mechanism that permits the developers to produce appropriate virtual reality software? There are some works that expose the relevance that a software toolkit makes to facilitate the development of a VR application [25].

This work proposed a method to design virtual reality games to be used for a healthcare project; to test the method, an application was developed and tested in a diabetes rehabilitation program. The method aims to improve the engagement of the patient with a virtual reality activity. To test this proposed method, a VR application was designed to help patients with type 2 diabetes to exercise, so patients can engage in physical activity in an attractive way. Furthermore, this work introduces a suggested test, selected from the most

common tests used in a virtual reality tool for health treatments, and could be a starting point for practitioners who wish to approach the study of new tools based on virtual or mixed reality to be used to improve health treatments that demand patient physical activity. Being able to apply this to patients recently diagnosed with chronic degenerative diseases, this study was conducted based on a program called home therapist for patients with type 2 diabetes mellitus and represents a challenge to support an emerging field, where patients need to be alone at home or with a few technical nursing assistants.

## 2. Materials and Methods

After identifying the health treatment features, a design process of a serious game for health can be synthesized into four steps (Figure 1), which involve a number of profiles that interact during the process: game designer, game developer, therapeutic specialists, physicians and patient.

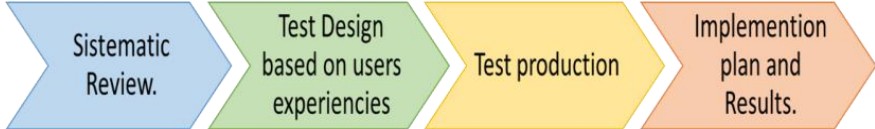

**Figure 1.** Four-step process to design serious games for health.

The first step requires a systematic review (both from scientific publications and from "grey literature"), of which the results are categorized into three critical aspects of the final application:

- User requirements;
- Game architecture;
- Health problem;

These three aspects dramatically affect the design of the application and the design of the tests for the application. Considering these elements will eliminate the waste of activities during the design.

The second step involves the design of the tests, based on user requirements, clinical tests related to the specific health problem, environment architecture, game mechanism, taxonomy, interactions, and interfaces. Figure 2 shows the most important items, as dimensions to consider when designing tests for virtual reality games for health. These dimensions help the designer to recognize what to test and how to plan such tests. Game developers and therapeutic specialists will present a list of tests to be conducted for each feature in the product. The first dimension corresponds to the clinical tests. This is the most relevant element at this stage, since clinical tests measure the actual impact of the virtual reality game on the health condition of the patient. The second one represents the physical activity target, and the third dimension is used to define the virtual environment. Health game mechanism taxonomy is identified at dimension four, and at the fifth dimension, interaction design is included; in addition, include the instructions on how to do the activities, and finally, the sixth one is used to define the interface design required according to the physical activity and its corresponding target.

The third step (test production) is related to the design of the test content, features, and guidelines. This step also comprises the design of the test bench for the user experience. During this stage, the health technical producer works with the plan and programming groups to ensure that everyone cooperates and works according to the physical activity test. The main job for them is to have the test steps followed by the therapist and programmers, ensuring that activity schedules are maintained and ensuring that the high-concept goals of the health game test are followed during the entire implementation process. When they both ran the building blocks, the production team optimized all parts of the game to work properly on the hardware and devices used to apply to the fitness goals.

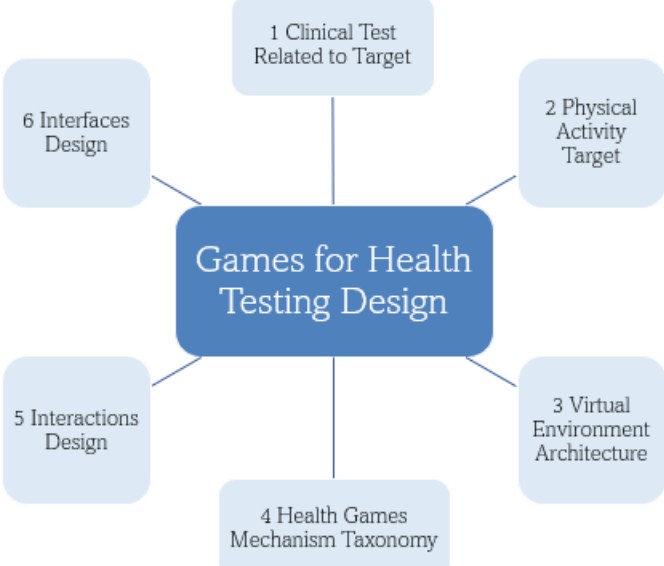

**Figure 2.** Critical items to consider when designing tests for games for health.

The fourth step comprises the implementation plan and collection of results. The timing for this stage heavily depends on the scope of the application, size and skills of the developers, and the quality of the outcomes of the previous steps.

For this paper, virtual reality tools were implemented to generate an application that engages the users. The result of the design step of the method was a three-exercise virtual reality application. The first two exercises focus on superior and inferior articulation movement. For the third task, a bike simulation is included as an additional exercise method for the patient.

Game architecture should be easy to replicate and close to the real task. The application is generated using a market-accessible HDM headset, so the project can be easily replicated in other places if needed. At the same time, as our method proposes, the game architecture should be close to the solution simulated by the game. Therefore, to simulate the bike minigame, a set of pedals was also installed for the users to feel like they are riding a bike.

Ethical Considerations: This protocol was registered and approved by the Research Committee of the Vice President of Health Sciences of the University of Monterrey (CIE), Ref.: 08052017-CI. All participants provided signed informed consent, and all data from patients were managed confidentially

### 3. Results

*3.1. Application Design: A Mixed Reality Video Game to Prevent the Physical Effects of Type 2 Diabetes Mellitus*

The four-step method described in Figure 1 can be applied to design a serious game that will be used as a tool for physical activity routines. The target users are recently diagnosed diabetic people between 21 and 60 years of age, who have access to a computer, virtual reality accessories, and an internet connection at their workplace or home. The application is aimed to motivate users to immerse themselves in the game, so that their interaction will not be static but physically active. The design process must also ensure that users will advance throughout the game, without interruptions produced by the incorrect use of the game's physical component, or by interactions with interfaces that are not oriented to cover the health target. The designed virtual reality application is shown in Figure 3.

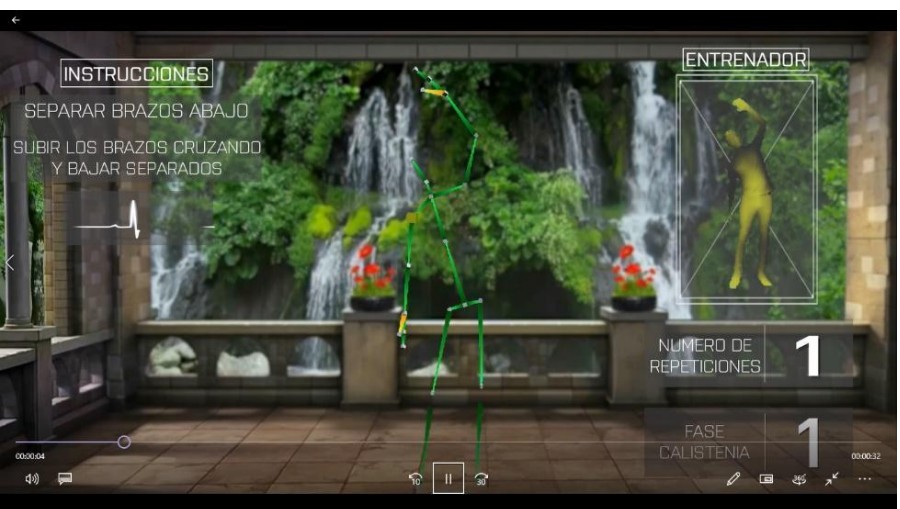

**Figure 3.** Implemented application related to diabetes.

The first step of the method led to the identification of a number of tests that had been applied within a videogame and to show the health questionnaire used together to test a physical activity game. Regarding medical tests for health, two instruments already validated, Cohen and EQ-5D-5L [26,27], were found that could be applied to Mexico and Spain. While Cohen is a test that identifies the level of stress when using the game, EQ-5D-5L is a test to describe the patient's health. The last one comprises five dimensions: mobility, self-care, usual-activities, pain/discomfort, and anxiety/depression. Each dimension has five levels: no problems, slight problems, moderate problems, severe problems, and extreme problems

During phase two, and apart from the two questionnaires mentioned above, two additional clinical tests were selected, VARK [28] and Barthel [29]. VARK is a questionnaire that clarifies how users work with information and their preferred learning style: visual, auditive, reader-writer, or kinesthetic. Barthel is a questionnaire to assess the ability to perform ten basic daily life activities. Hence, these tests offer a quantitative estimate of the degree of independence of the patient that can be used to improve the game interaction

Additionally, three tests were applied:

- Functionality tests recognize deviations from functional requirements
- Usability tests reflect the user experience with an interactive system to achieve a goal performing a series of specific tasks.
- Playability tests a series of attributes that allow us to characterize the experiences of a player before using the video game, i.e., the set of factors that satisfies the patient while doing a rehabilitation activity.

To select elements for the clinical test, a PICO tool for research questions was used. The main questions of this research are: what are the requirements to design a virtual reality application that will be used in a health area and how is that a special test designed for virtual reality to be used based on physical activity contributes to advances based on better patient acceptance and consequently fulfill the health game target?

The third step produced a test case that could be applied in a physical activity video game that uses virtual reality tools. The combined test can help prevent some physical impacts while the patient interacts with video game devices. To assess user experience, tools such as a controls interface, 3D models, Mix Reality visors, and game controls were used, which permitted a more real experience while the user interacts with game scenes. At this step, we needed to know the emotional effect on the patients. We selected six emotion items: stress, engagement, interest, excitement, focus, and relaxation. A wireless headset based on electroencephalography (EEG) was used to read data that represent the emotions

(Figure 4). Together with the EEG test, we selected four user experience items: instructional guide, clarity, interface experience, and difficulty experience.

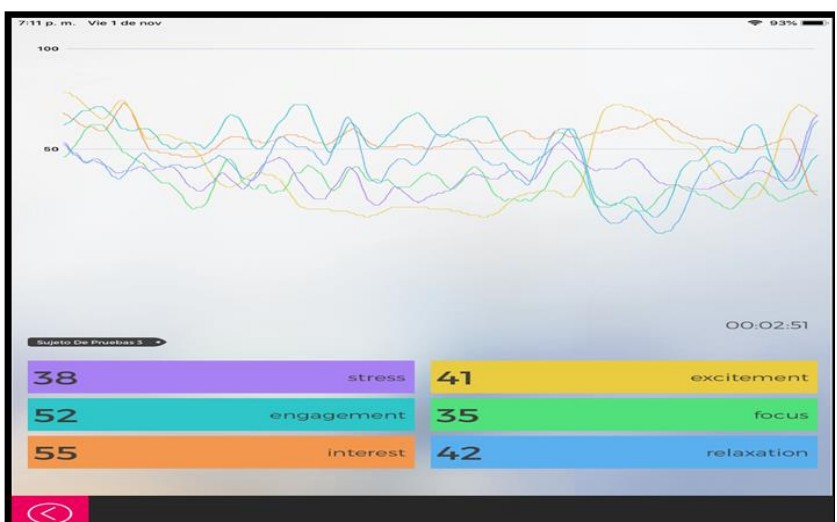

**Figure 4.** EEG used to collect data on real emotions.

Eye tracker analysis was conducted during the usability test (Figure 5), using software developed by the Eye Tracker company. Some of its functions are generating images, videos, and screen recordings to be analyzed. Hence, the researchers were able to identify a relationship with some emotive data recollected, as focus and engagement with and interest in the activities.

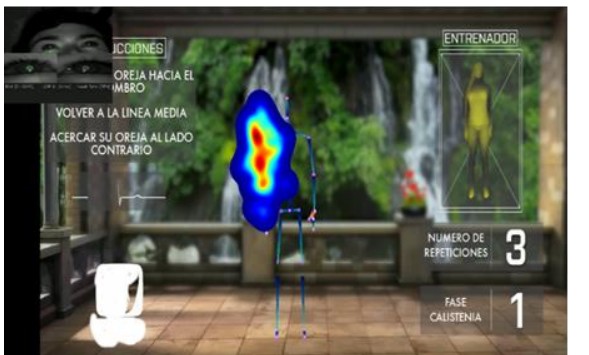 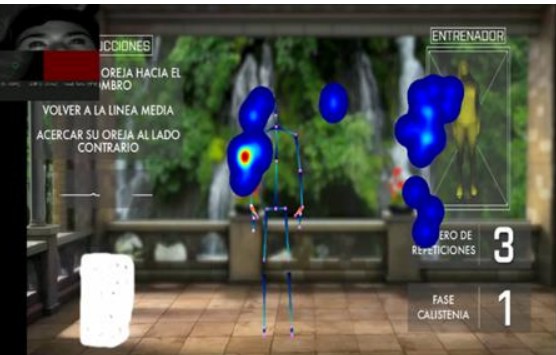

**Figure 5.** Gazepoint Analysis, eye tracker recording user game vision.

The whole process for applying the described method took eleven months: five months to develop interactive experiences; four months to test the T2DM physical activity game; and finally two months to validate and improve the results.

### 3.2. Case Study: Results from the Interaction with the Videogame

The design of the test started with 30 participants, who were evaluated under normal working-hour conditions. Twenty-seven patients completed all evaluations; the rest, three participants, were excluded due to incomplete data. Table 1 shows the sociodemographic characteristics of the participants. The age of patients covered a broad range from 24 to 77 years (median: 55 years). Due to the size of the sample, we described the main outcomes as proportions.

**Table 1.** Sociodemographic characteristics.

| Characteristics | | |
|---|---|---|
| Age, mean (SD) | | 53.85 (12.38) |
| Age group, *n* (%) | Under 55 | 9 (33.3%) |
| | Over 55 | 18 (66.7%) |
| Genre, *n* (%) | Male | 5 (18.5%) |
| | Female | 22 (81.5%) |
| Civil status, *n* (%) | Single | 2 (7.4%) |
| | Married | 17 (63.0%) |
| | Divorced | 4 (14.8%) |
| | Free union | 2 (7.4%) |
| | Widow | 2 (7.4%) |

Table 2 shows the main results of the clinical tests. According to the Cohen Test, the users of the application had a low level of stress at 66.7%. The application of the VARK questionnaire showed that most of the users of the video game were mainly visual and kinesthetic (29.6% each). Furthermore, 70% showed good mobility, 88.8% showed good personal care, and 86% perceived a good state of health, according to the EQ-5D scale. Finally, the results of the Barthel test indicated that 70.4% were completely independent and only 29.6% reported low dependence.

**Table 2.** Results from the clinical tests.

| Clinical Test | Scale/Dimension | Proportion (*n*) |
|---|---|---|
| Cohen | Scores ranging from 0 to 13 (low stress) | 66.7% (18) |
| | Scores ranging from 14 to 26 (moderate stress) | 25.9% (7) |
| | Scores ranging from 27 to 40 (high perceived stress) | 7.4% (2) |
| VARK | Visual | 29.6% (8) |
| | Auditive | 18.5% (5) |
| | Readers | 22.3% (6) |
| | Kinesthetic | 29.6% (8) |
| **Clinical Test** | **Scale/Dimension** | **%Without Affection (*n*)** |
| EQ-5D Test | Mobility | 70.3% (19) |
| | Personal Care | 88.8% (24) |
| | Daily activities | 85.1% (23) |
| | Pain/discomfort | 40.1% (11) |
| | Anxiety/depression | 66.7% (18) |
| | Perceived health status | 86.2% (23) |
| Barthel test | <20 Full dependence | 0.0% (0) |
| | 20–35 High dependence | 0.0% (0) |
| | 40–55 Moderate dependence | 0.0% (0) |
| | ≥60 Low dependence | 29.6% (8) |
| | 100 Independent | 70.4% (19) |

After checking the normality of the data using the del EQ-5D test with the Kolmogorov-Smirnov method, it was clear that most of the parameters did not follow a normal distribu-

tion. In order to recognize whether young or older people had differences between EQ-5D activities, samples were divided into two groups by age according to the group median (<55; ≥55).

Although differences in base conditions between groups would be expected, no significant differences were found when using the U-Mann Whitney test.

All patients completed the health goal of performing 30 min of physical activity with a Monday–Friday frequency during 4 months of follow-up. All of them completed the usability tests designed for this purpose. Table 3 shows the outcomes of the usability test recorder while watching the video: in general, the perception of the users was excellent or good towards the usability of the video. The results also led to the identification of those items that offered the best opportunities to improve the overall experience, such as the waiting time.

**Table 3.** Usability table.

| Questions | Value | Proportion (*n*) |
|---|---|---|
| Did the video give you the necessary instructions to know what actions to take? | **Yes** <br> No | **100% (27)** <br> 0% (0) |
| How do you rate the instructions provided to interact with the video? | **Excellent** <br> Very Good <br> Good <br> Medium | **37% (10)** <br> 26% (7) <br> 37% (10) <br> 0% (0) |
| How do you rate your experience and interaction with the interface? | **Excellent** <br> Very Good <br> Good <br> Medium | **33.3% (9)** <br> 29.7% (8) <br> 33.3% (9) <br> 3.7% (1) |
| How do you rate the waiting times during your interaction with the exercise video? | Excellent <br> Very Good <br> **Good** <br> Medium | 29.7% (8) <br> 22.2% (6) <br> **44.4% (12)** <br> 3.7% (1) |
| How do you rate the handling of the steps to follow the exercise routine? | **Excellent** <br> Very Good <br> Good <br> Medium | **37% (10)** <br> 26% (7) <br> 37% (10) <br> 0% (0) |
| What moment of your interaction with the video was the most memorable? | **All** <br> Walking <br> Warm-up <br> Stretching | **38.5% (10)** <br> 26.7% (7) <br> 23.1% (6) <br> 7.7% (2) |
| Did you have difficulty while interacting with the exercise video? | Yes <br> **No** | 3.7% (1) <br> **96.3% (26)** |
| Difficulties in | **Stretching** | **100% (1)** |

Only 27 patients (5 male, 22 female) completed the test for emotive reactions and user experience (Table 4). These evaluations were also analyzed with non-parametric statistics, the U Man Witney test, because of the non-normal distribution of the data. The results showed that the emotion with fewer results was stress, and due to the 100-point metric of stress expected, the mean was 36.48. The emotions level is represented using a scale from 0 to 100, where <50 represents lower stress emotions and >50 represents a major stress level.

Of the dimensions analyzed, only stress (*p*-value = 0.03) showed significant differences between the age groups. Older people had higher stress levels (median value = 12.77 than younger ones (median value = 6.58). A comparison of other dimensions showed no significant differences.

**Table 4.** Emotive reactions and user experience while the video game is developed, mean and standard deviations (SD).

|  | Mean (SD) |
|---|---|
| Stress | 36.48 (5.73) |
| Engagement | 48.05 (16.18) |
| Interest | 54.86 (5.73) |
| Excitement | 42.90 (5.50) |
| Focus | 34.52 (6.46) |
| Relaxation | 34.52 (12.56) |

With regards to finishing the activities that are the key components of the physical treatment, the group over 55 years of age showed a better average based on reaching the health target while using the virtual reality tool.

The users of the virtual reality tool stated that they had enough instructions during their interaction with the videogame. Most of them felt that the contents were clear, had a good interface experience, and reached their health goals. Moreover, most of them (90.5%) did not experience difficulties during its use (Table 5), and there were no significant differences when age groups were compared (data not shown).

**Table 5.** Outcomes of user experience in proportions according to the Likert scale.

| User Experience | Scale | Proportion (*n*) |
|---|---|---|
| Enough instructions | Always | 57.1% (12) |
|  | Almost Always | 28.6% (6) |
|  | Sometimes | 14.3% (3) |
|  | Almost never | 0% (0) |
| Clarity | Always | 4.8% (1) |
|  | Almost Always | 42.9% (9) |
|  | Sometimes | 38.1% (8) |
|  | Almost never | 14.3% (3) |
| Interface Exp | Always | 4.8% (1) |
|  | Almost Always | 28.6% (6) |
|  | Sometimes | 61.9% (13) |
|  | Almost never | 4.8% (1) |
| Health Target | Always | 9.5% (2) |
|  | Almost always | 66.7% (14) |
|  | Sometimes | 14.3% (3) |
|  | Almost never | 9.5% (2) |
| Difficulties | No | 90.5% (19) |
|  | Yes | 9.5% (2) |

## 4. Discussion

As seen in the introduction, there are different approaches to the design of a tool based on virtual reality games to be used in health treatments. We have proposed a method for which the objective is to facilitate this entire design process. It is not only intended to make the process simpler but also to integrate the use of validated tools that increase the credibility and robustness of the process and allow greater guarantees that the developed tool will effectively meet its health objectives.

By relying on these clinical tests, the development process of the tool can be linked to evidence-based medicine [30]. For example, in the case study presented, the development process is supported by tests from the field of rehabilitation.

Among the existing approaches that can be considered similar to our work, the most remarkable is that of Mayer et al. [19]. Our proposal goes further than Mayer's and includes, among the most important design elements, an interaction element where patients are informed before starting of how to follow game instructions, how to be physically prepared according to their own health requirements, and how and when to record their vital signs. As seen in the results (Table 5); this integration is highly valued by users.

The results of this work show that the proposed method improves usability and user experience. Results were also obtained with respect to the "emotional reactions" of the users, showing that focus and engagement promotes the patient's motivation to do the activities and therefore achieve the goal of treating physical activity in patients with T2DM. The results of the study have shown that people who applied this case have had a remarkable improvement in following the re-habilitation activities promoted by the videogame. Additionally, to the results presented, they were asked, in a private form, whether they would recommend using video games for their rehabilitation and most of them agreed with that statement.

It was not the purpose of this work to demonstrate the clinical effectiveness of the tool presented as an example. A possible evaluation, with a randomized clinical trial, of the effectiveness of the tool remains a future goal.

An additional consideration when developing a health care virtual reality solution is the scope of the proposal. The features included in a proposal are based on the needs of the patients. However, due to the complexity of the disease, there are different approaches that depending on time, could not be included in the design. Design should prioritize the features that would help the user the most. Nauta and Spil [7] proposed a modular design, which they describe as a base application that can be later adapted to the user needs. From patient to patient, treatment may have small variations. A virtual reality design scope should be able to adapt to the main needs of the users and could be applied for other health issues [31].

Although the case study presented in this article refers to a virtual reality tool, the method we propose could also be applied to any extended reality product that involves all real-and-virtual combined environments and human–machine interactions, generated by computer technologies and wearables devices.

## 5. Conclusions

Presented here is a method to improve the design of virtual reality videogames for health, based on scientific evidence and comprehensive testing. This leads to the conclusion that after testing the model in the health-videogame, depending on how they are executed, the elements included in the method help to reach the purpose despite different purposes.

The results could lead to new ways to design products, based on this method, which support activities using mixed reality concepts. It could also be observed that using validated instruments for health tests reflects a benefit based on the amount of time needed to complete the videogame; this finding could be a subject for future research.

The use of virtual reality as a tool to develop environments has the advantage of generating engagement. In the present work, most of the users were engaged in the use of the diabetes game application. Is important to highlight that evaluating the health benefits of the tool would require a further analysis. However, with enough instructions provided, the virtual experience was of interest for the users that participated in testing. Most users indicated that the interface was easy to use. At the same time, user experience tests showed that the users felt that the interaction was acceptable. An effective virtual reality design is a useful tool in order to generate engagement from the user.

**Author Contributions:** Conceptualization, L.N.-T. and F.G.S.; methodology, L.N.-T.; software, L.N.-T., original draft preparation and writing, L.N.-T. and F.G.S.; project administration, I.C.R. All authors have read and agreed to the published version of the manuscript.

**Funding:** This research received no external funding.

**Informed Consent Statement:** Not applicable.

**Data Availability Statement:** Not applicable.

**Conflicts of Interest:** The authors declare no conflict of interest.

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
