# Peer review of "A Method to Improve the Design of Virtual Reality Games in Healthcare Applied to Increase Physical Activity in Patients with Type 2 Diabetes Mellitus"

_applsci, doi:10.3390/app13010050_

Round 1
Reviewer 1 Report
It is not clear in the text whether, based on the described method, the authors create a game and apply the methods, describing the results of the application of the game on 27 participants? If so, it might be good to give more information about the game, not just the method on which it was constructed, as the results relate to its application, or to refer to another source where the game is described.
214 Case study: results from the interaction with the videogame – I think “case study” is not a correct term
248 - Only 27 patients (6 male, 15 female) – needs correction
I would recommend to indicate whether the ethical requirements are met
Author Response
Please see the attachment
Point1: A more detail related to the game was present in a last paragraph of method section
Point2:The “case study” term was replaced by Application design, for better understanding.
Point3: The typing error on total male and female has been updated (5 male,22 female).
Point4: A paragraph that describes the ethical requirements has been included at the last part of section 2, materials and method.

Reviewer 2 Report
Thank you for this contribution. This is an interesting and timely manuscript. A good piece of work has been done
Author Response
Reviewer 2 has No points to respond
Reviewer 3 Report
E-health related implementation of VR has been so far very impressive, particularly in well-being and CBT (Cognitivistic-Behavioral Therapy) areas promoting innovative uses of highly immersive media. The authors’ original research, however has little relevance to VR technology. The authors’ original approach, derived from clinical and usability studies/ analysis (in fact explotinig retention and adherence od therapies) can be used to trace performance, verify completion of the tasks and improve organization of (virtual) therapy. Moreover, Sulbaran and Baker showed that learners usually enjoy VR training more than other traditional training methods and that they can retain knowledge gained from VR training longer than that gained using other methods [Sulbaran T, Baker NC (2000) Enhancing engineering education through distributed virtual reality. In: ASEE/IEEE frontiers in education conference. Kansas City, MO, pp 3–18]. Recent studies carried by Baukal and Ausburn show that the retention rates for VR learning reached over 75 % comparing to 10% for reading and less than 50% for lecture style learning [Baukal, C. E., Ausburn, F. B., & Ausburn, L. J. (2013). A Proposed Multimedia Cone of Abstraction: Updating a Classic Instructional Design Theory. Journal Of Educational Technology, 9(4), 15-24].
Summing up, the paper is of high quality but mistargeted...
The paper is written in a very clear style that cannot cause ambiguities. The described method is backed up by appropriate and convincing results. However I think that the paper in the current form is not suitable for publication, strictly VR-related research is not the main objective of the paper. I suggest eiher submittin the paper to e-health related journals re-profiling the paper title and contents, putting stress on authors’ genuine achievement in DTX (Digital Therapeutics) and e-health systems design/verification. Title, abstract, Chapter 1 and Conclusions need to be rewritten, then.
Author Response
Point1: To clarify that this work is related to VR research as main objective and the eHealth objective is a secondary objective, it will include a paragraph explain the virtual reality tool developed following the method to designed virtual reality games that is proposed at this paper.
Point2: Does the introduction provide sufficient background and include all relevant references? Four references are included, to improve the background related with VR.
Point 3: Are the conclusions supported by the results?, Response 3: To improve conclusions , it has been rewritten including the importance that reflect the result on the user experience when using a virtual reality tool designed for a specific needs.

Round 2
Reviewer 3 Report
The paper has been improved. Maybe it needs some minor language correction in the updated sections (e.g. "The result of the design step of the method was a three exercises virtual reality application. " I guess it means vr app featuring three experiences...).